# Impact of communication anxiety on L2 WTC of middle school students: Mediating effects of growth language mindset and language learning motivation

**Juan Wang**[1,2]*, **Tongquan Zhou**[3], **Cunying Fan**[1]

**1** Department of College English Teaching, Qufu Normal University, Qufu, Shandong, China, **2** College of Chinese Language and Literature, Qufu Normal University, Qufu, Shandong, China, **3** School of Foreign Languages, Southeast University, Nanjing, Jiangsu, China

* wangjuan2003@qfnu.edu.cn

**Data Availability Statement:** All relevant data are within the manuscript and its supporting information files.

## Abstract

Previous research has shown a connection between communication anxiety and willingness to communicate (WTC) among English as a foreign/second language (L2) learners. Nonetheless, the potential mediating roles of learners' beliefs like growth language mindset and language learning motivation have not been thoroughly investigated, particularly in the context of middle school language learners. This study aimed to explore the relationship between communication anxiety and L2 WTC by considering the mediating roles of growth language mindset and language learning motivation. To achieve this goal, an online survey was administered to 847 participants from five middle schools in eastern China. Structural equation modeling was employed to analyze the collected data. The findings revealed that L2 WTC was negatively impacted by communication anxiety and growth language mindset, while positively influenced by language learning motivation. Additionally, communication anxiety had a negative effect on both growth language mindset and language learning motivation, whereas growth language mindset exerted a positive effect on language learning motivation. Moreover, either individually or synergistically, growth language mindset and language learning motivation played mediating roles in the relationship between communication anxiety and L2 WTC. These research findings have significant implications for understanding the interrelationship among the variables, offering an innovative perspective on the mediating effects of growth language mindset and language learning motivation on communication anxiety and WTC among secondary school students. Consequently, this study provides valuable insights for language learning and instruction among middle school students and teachers.

## Introduction

In the process of learning a foreign/second language (L2), some learners may actively seek out opportunities to communicate in L2, whereas others may choose to remain silent and avoid

**Funding:** This work was supported by the Teaching Innovation Program from Qufu Normal University (grant number:22jg41), and the funder had no role in study design, data collection and analysis, decision to publish, or preparation of the manuscript.".

**Competing interests:** The authors have declared that no competing interests exist.

communication despite possessing strong language proficiency and recognizing the value of communication. To predict learners' propensity for communication in an L2, MacIntyre et al. [1] proposed and conceptualized the concept L2 willingness to communicate (L2 WTC). They posited that L2 WTC referred to learners' readiness to engage in discourse at a specific time with certain individuals using L2, and representing the final psychological stage before actual communication behavior. Therefore, it is crucial for L2 instructors to stimulate learner' WTC, and for researchers to investigate the various factors that affect L2 WTC.

Studies have shown that L2 WTC is influenced by multiple factors, including both individual differences—such as communication anxiety, growth language mindset, and language learning motivation as evidenced by recent research by Sadoughi & Hejazi [2] and situational factors like classroom environment, and topic familiarity, among others [3]. Among the factors, communication anxiety refers to the feelings of worry, fear, or discomfort that individuals experienced when communicating in L2, which may lead to a decline in language performance [4]. Growth language mindset, contrasting with fixed language mindset, is the belief that language abilities are malleable and improvable through practice and hard work [5]. This motivational resource serves as a guide for learners to concentrate on the processes of improvement and learning. Language learning motivation is the combination of effort, desire, and positive attitudes towards learning [6]. As posited by Li et al. [7], students who are highly motivated and possess a strong inclination to learn are generally observed to exhibit a higher level of engagement than their less motivated peers. Every single one of these factors has the potential to impact L2 WTC of individuals and also interrelations existed among some of these factors. Nonetheless, despite their significance, few studies have examined their intertwined relations within a single model. With the emotional/affective turn that recognizes the significance of Positive Psychology in L2 learning, a new avenue of research into the impact of learner emotions, particularly positive emotions and affect on L2 WTC is gaining momentum. Currently, the majority of relevant literature delved into the relationship between two among the three factors like relationship between mindset and anxiety [8–10], motivation and WTC [11–14], mindset and WTC [15, 16], anxiety and WTC [17]. And the associations among three of them: for instance, motivation, growth mindset and WTC [18, 19], emotions, mindset and WTC [20], anxiety, language learning motivation and L2 WTC [21] were also examined. However, the majority of these studies have primarily focused on university students, neglecting middle school students who are at a critical period characterized with significant psycho-social development [22]. The experiences encountered during this stage can have a profound effect on the motivational outcomes of students, given the diverse biological, cognitive, social, and educational transformations they are undergoing. Furthermore, the early stages of adolescence represent a crucial period of development during which students are inclined to engage in a rigorous self-assessment process with fixed mindset [23]. Additionally, L2 WTC of students in Asian society is typically deficient due to the emphasis on examining receptive skills (e.g., reading and listening) in English courses, which have been condemned for failing to prioritize communication in the target language [24].

In light of the developmental factors of middle school students and the features of English courses in China, it is essential to explore how communication anxiety, language learning motivation, and growth language mindset, which are interconnected yet distinct constructs, impact L2 WTC of middle school students. This study examines a mediation model that incorporates communication anxiety, growth language mindset, language learning motivation, and L2 WTC. It is expected to highlight the role of learner-related factors in cultivating L2 WTC, offering practical insights for the development of educational strategies aimed at enhancing L2 WTC.

## Literature review and hypotheses development

### Willingness to communicate in L2

L2 WTC is considered a multifaceted construct that encompasses numerous linguistic, affective, social-psychological, and communicative influences [1]. L2 WTC fosters increased learning opportunities and serves as a primary goal of language instruction. However, attaining a state of L2 WTC is a complex process influenced by numerous factors, as described by MacIntyre et al. [1] in their pyramid-shaped L2 WTC model. These factors vary from enduring influences, such as personality traits, to situational influences, including state anxiety, task characteristics and interpersonal factors. The interplay between psychological factors like anxiety, excitement, and security, and situational variables such as topic, interlocutors, and conversational context has also been found to exert a significant impact on L2 WTC [25]. Furthermore, various individual difference variables, including language learning beliefs and motivation, have also been suggested to impact WTC [12, 21, 26].

Research on L2 WTC has been particularly robust in East Asian EFL settings, as exemplified by Yashima's [12] work serving as a notable example. The research demonstrated that ideal L2 selves, which encompassing learners' aspirations and their envisioned future linguistic identities, significantly correlated with WTC in and out of the classroom in English among Japanese high school students. Peng [13] conducted a study in China involving 1,013 undergraduates, revealing that L2 WTC consists of WTC inside and outside the classroom and WTC inside the classroom was affected by L2 anxiety. Ma et al. [27] undertook a qualitative investigation into the factors that contribute to L2 WTC among four Chinese postgraduates in English-mediated academic learning context. The study identified a range of influential variables at individual, environmental, social-cultural levels that significantly shape L2 WTC experiences. Among the individual factors, attitude and motivation were found to affect learners' L2 WTC [18, 21, 26]. The most recurrent affective factor is anxiety, which usually arises when students worry about their own performance and potential negative judgment from peer students [28].

Upon reflection, prior investigations into L2 WTC have laid a substantial theoretical basis for the current study. However, there has been insufficient focus examining the associations among WTC, emotions, and beliefs. Additionally, our study features middle school students, providing a novel perspective for examining the relationships among WTC, emotional factors, and personal beliefs.

### Relationship between communication anxiety and L2 WTC

Anxiety, one of the dominant negative emotions, has been the subject of extensive research in SLA, with a particular focus on communication anxiety. The reason why we choose communication anxiety as a variable to investigate is because we want to ascertain, under the influence of the positive beliefs such as language learning motivation, growth language mindset, whether we could mitigate the adverse effects of communication anxiety on L2 WTC. As is known, communication anxiety arises in connection with real or anticipated communication with others and is a common occurrence among language learners at various stages of their learning journey [28].

Previous research identified that communication anxiety has been a significant negative predictor of an individual's L2 WTC [29, 30]. Anxious learners experience concerns related to potential failure and negative evaluation by others, which may divert and reduce their mental resources and concentration, leading to a significant decrease in their L2 WTC and an underestimate of their linguistic competence. There is a strong association between high levels of communication anxiety and learners' reluctance to communicate in English [31]. Therefore, we hypothesize that communication anxiety is a negative predictor of L2 WTC:

*H1*: *Communication anxiety negatively predicts L2 WTC of Chinese middle school students.*

## Relationship between growth language mindset and L2 WTC

The concept of mindset is firmly entrenched within the social-cognitive framework of achievement motivation, which proposes that individuals harbor implicit or explicit beliefs or theories about their attributes, such as cognitive aptitudes and dispositions. Research on growth language mindset has become a rapidly expanding area in L2 learning and teaching, as evidenced by recent research [10, 32]. Learners with this mindset are more likely to invest time and effort into their language learning, focusing on learning goals rather than performance [23]. Furthermore, they place greater emphasis on the learning process rather than their performance and are more inclined to take responsibility for their learning and foster autonomy [5]. They view mistakes as opportunities for growth, employ self-regulatory strategies, and experience more positive emotions. According to Lou and Noels [10] these learners are willing to put in more effort and practice, which could enhance their engagement in L2 communication.

According to recent research, growth language mindset has a positive impact on L2 WTC. Zarrinabadi et al. [15] conducted a study involving 106 first and second-year undergraduates in Iran to investigate the relationship between growth language mindset and WTC. Their findings demonstrated a positive correlation between the two factors. Additionally, they found that growth language mindset acted as a mediator for autonomy support, which subsequently impacted WTC. Another study by Wang et al. [20] proposed that although WTC and language mindset were not directly related, the latter had an indirect effect on the former through mediators such as boredom, enjoyment, and pride. Previous study has yielded mixed results, therefore in his more recent study, Zarrianabadi [33] advocated that more weight should be given to the significance of growth mindset and invited avid researchers to examine whether it could be a facilitator of WTC in EFL contexts. In response to his call, our study concurs with Zarrinabadi et al. [15] that growth language mindset would significantly influence WTC directly.

In light of the aforementioned findings, we posit the following hypothesis:

*H2*: *Growth language mindset positively predicts L2 WTC of Chinese middle school students.*

## Relationship between language learning motivation and L2 WTC

Motivation is a crucial factor in L2 learning, which encompasses effort, desire, and positive attitudes towards language learning [6]. Extensive research has been conducted on the relationship between language learning motivation and L2 WTC, and it has been established that there exists a direct positive correlation between motivation and L2 WTC [34–37].

Hashimoto [34] further supported this finding by investigating the predictive power of affective factors, particularly motivation, on WTC of Japanese EFL learners in a classroom setting and revealed a positive association between WTC and motivation.

Alrabai [36] explored the complex interplay between motivation and WTC in the context of EFL learning. The study confirms that motivation has a direct positive effect on learners' WTC. Additionally, the analysis revealed that motivation emerged as the most salient predictor of L2WTC.

Zhang and Dai [37] further highlighted the positive links between learning motivation and L2 WTC on Chinse EMI (English as the Medium of Instruction) students, which extends the understanding of how motivation can foster a proactive approach to L2 WTC in a new educational context.

Based on these findings, we propose the following hypothesis:

*H3*: *Language learning motivation positively predicts L2 WTC of Chinese middle school students.*

## Relationship between communication anxiety and growth language mindset

Anxiety is a well-researched emotion in L2 study. It is characterized by negative emotional responses and worry that arise during the process of acquiring or using a new language. Communication anxiety arises during actual or anticipated interaction with one or more other individuals. Individuals who suffer from communication anxiety tend to exhibit a heightened sensitivity towards social rejection [9], resulting in negative experiences and limited use of L2. This, in turn, reinforces their low perceptions of their L2 competence, leading to the adoption of fixed mindset. This fixed mindset is characterized by the belief that language skills cannot be improved, which not only hinders the development of growth language mindset, but also poses a significant barrier to successful L2 acquisition. According to a recent investigation carried out by Lou and Noels [10], migrant students enrolling in a post-secondary institution experience anxiety when the language of instruction is not their native tongue, particularly when utilizing the language in their day-to-day social interactions. This anxiety can lead to a reluctance to employ the target language. With the passage of time, their performance will be greatly reduced. Such a decline might reinforce the belief that linguistic capability cannot be changed, thereby fostering a fixed mindset.

Drawing from the research, we put forward the following hypothesis:

*H4*: *Communication anxiety negatively predicts growth language mindset of Chinese middle school students.*

## Relationship between communication anxiety and language learning motivation

Scenarios that call for oral communication in L2 can generate substantial amounts of anxiety within learners [28]. Communication anxiety has been found to have a negative impact on language learning motivation [34]. When learners experience high levels of anxiety related to communicating in the target language, they may exhibit more passive behaviors, feel reluctant to take risks in expressing themselves, fear making mistakes, lack self-confidence, be concerned about being less competent than their peers, and have low levels of motivation for language learning [34–36].

Liu and Zhang's [38] study of 1697 Chinese university students showed a significant negative association between foreign language anxiety and English learning motivation. In the same vein, Kruk's [39] research on advanced English learners revealed a negative relationship between motivation and anxiety. Furthermore, Salayo and Amarles [40] found that language learning anxiety also had a negative impact on motivation in 39 Grade 3 Filipino learners of English. Therefore, after the literature review, we proposed that communication anxiety, as one kind of language learning anxiety, has a negative relationship with language learning motivation.

*H5*: *Communication anxiety negatively predicts language learning motivation of Chinese middle school students.*

## Hypothesized model

Drawing from the theoretical underpinnings and literature reviewed, a hypothesized model was developed by integrating four latent variables: the negative emotion of communication

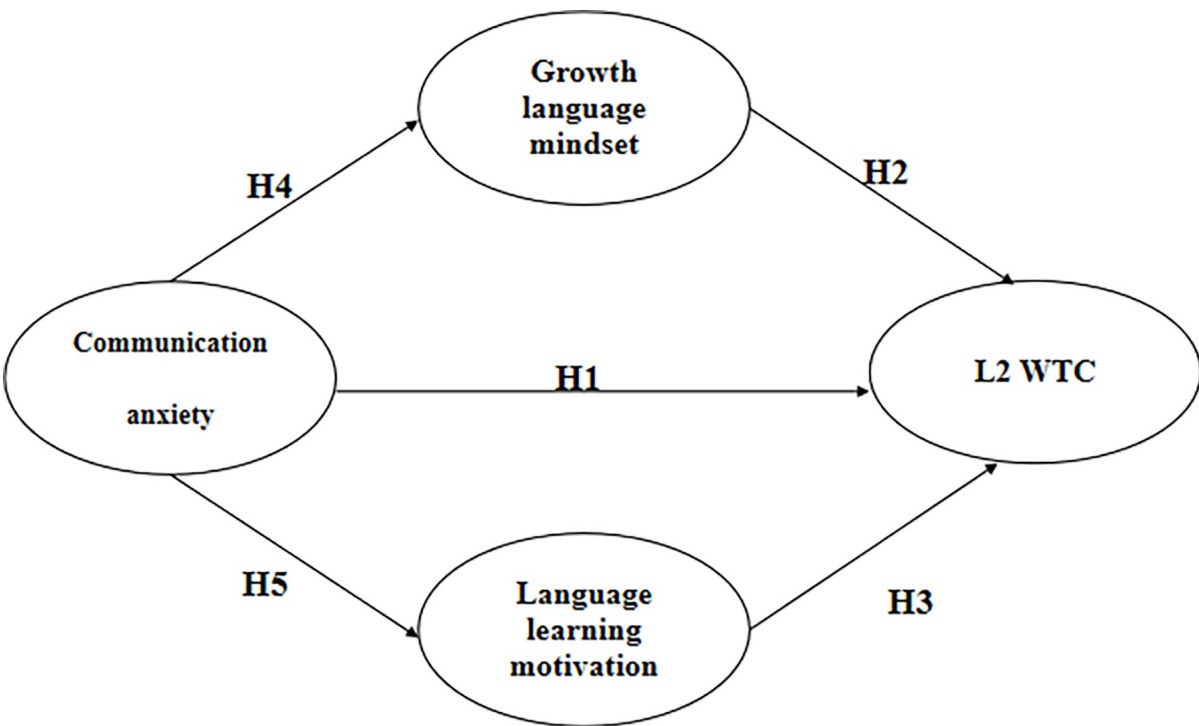

**Fig 1. The hypothesized model of association between communication anxiety and willingness to communicate in L2 under consideration of growth language mindset and language learning motivation.**

anxiety, and the affective variable of language learning motivation and growth language mindset and learners' L2 WTC as an outcome variable among Chinese middle school EFL students. Model specifications were constructed based on the theoretical assumptions grounded in the fields of positive psychology as well as findings of L2 WTC empirical research. For example, communication anxiety was speculated to be negatively associated to growth language mindset [9] and the positive affect of motivation [29–31], and motivation, on the other hand, was anticipated to influence L2WTC directly, based on empirical evidence [34, 36, 37] and growth language mindset directly influence L2 WTC [15, 20]. According to the previous research, we proposed the following conceptual model in Fig 1 with growth language mindset and language learning motivation as potential mediators.

H6: *Growth language mindset mediates the relationship between communication anxiety and L2 WTC of Chinese middle school students.*

H7: *Language learning motivation mediates the relationship between communication anxiety and L2 WTC of Chinese middle school students.*

## Method

### Participants

This study involved a sample of 847 middle school students from China, aged between 12 and 15 years ($M = 13.412$, $SD = 1.004$). The sample was equally divided between males (49.2%) and females (50.8%), indicating a balanced gender distribution. The participants were in different grades, with 39.1% in 7th grade, 27.1% in 8th grade, and 33.8% in 9th grade. In terms of their

English learning experience, 22.4% has been learning English for three to four years, 43.6% for five to six years, and 34.0% for seven to eight years.

## Instrument

A survey instrument was developed to collect data for the purpose of testing hypotheses related to communication anxiety, growth language mindset, language learning motivation, and L2 WTC. The instrument was created by adapting and modifying validated scales from relevant literature to suit the learning environment of Chinese middle school students:(1) Items were rephrased to reflect cultural norms and experiences specific to middle school students;(2) Language was simplified to ensure it is accessible to youngsters; (3) The number of items was reduced as well as structure simplified to reduce the cognitive load. The English questionnaire was translated into Chinese by the first writer and the translated questionnaire items were subsequently submitted to three linguistic experts for validation to ensure their accuracy. Based on the feedback provided by the experts, the language of the translated items underwent further refinement. The first section of the questionnaire gathered demographic information such as gender, age, grades, and years of English language learning. The second section comprised 25 items that measured communication anxiety (9 items), growth language mindset (4 items), language learning motivation (7 items), and L2 WTC (5 items).For the detailed items, please see S1 File.

To assess communication anxiety, we modified nine items from Horwitz et al. [41] to better suit our context. Participants were directed to rate their degree of anxiety when communicating in English (e.g., "I am afraid that the other students will laugh at me when I speak English.").

The growth language mindset scale consisted of four items extracted and modified from Lou & Noels [42]. These items represent the participants' beliefs in the malleability of their language proficiency (e.g., "I can always substantially change my English language ability").

To evaluate respondents' language learning motivation, we utilized seven items from Liu [43]. The purpose of these items was to evaluate the motivational attributes of L2 learners as they pertain to their English language acquisition (e.g., "I usually spend lots of time studying English").

The evaluation of participants' WTC in English was carried out by using a set of five items. These items were adapted and modified from Baghaei & Dourakhshan [44]. The items aimed to assess the participants' WTC in English. (e.g., "If I encountered non-native speakers of English (Japanese, Korean, etc.), I would talk to them in English").

All the subscales assessing communication anxiety, language learning motivation, growth language mindset and L2 WTC were presented in a 5-point Likert scale format, with "1" indicating a strong sense of disagreement and "5" signifying a strong sense of agreement.

## Data collection

This study was structured to guarantee the confidentiality and anonymity of all participants involved. Before initiating data collection, ethics approval was secured from the Institutional Review Board (IRB) of the corresponding author's university: Qufu Normal University (approval number:2024–067) on Feb. 1, 2024. The research was conducted in compliance with the guidelines established by IRB of Qufu Normal University and the World Medical Association Declaration of Helsinki. Informed consent was obtained electronically from all participants. Additionally, permission was sought from their parents, and teachers. Furthermore, the participants were at liberty to withdraw from the study at any point without being obligated to provide an explanation for their choice.

Once we had obtained authorization from the relevant stakeholders, a comprehensive survey was initiated among student populations across five middle school in eastern China, commencing on the 1st of February, 2024. The survey was designed to span a duration of four weeks. The participants completed the verified and translated Chinese version of the questionnaire via an online platform (https://www.wjx.cn). The link to the platform was distributed to L2 learners by their English teachers through emails, QQ, and WeChat. The survey was available for two weeks and received 873 responses from middle school students. After eliminating cases with missing values and outliers, 847 valid cases (S1 Data) were retained for data analysis. Throughout the entire research process, the authors adhered to the ethical standards established by their university.

## Data analysis

We employed SPSS 26 to input our data and compute descriptive statistics. After we conducted descriptive statistics, a two-step analysis was performed on the data. Firstly, confirmatory factor analysis (CFA) was utilized to assess the measurement models for the four constructs. Secondly, structural equation modeling (SEM) was employed to scrutinize the structural model. Mplus 8.3 was utilized for both CFA and SEM procedures.

To test the measurement model, we analyzed the reliability and validity of the survey through the examination of individual item reliability, internal consistency reliability, convergent validity, and discriminant validity. The goodness-of-fit between the model and the data was assessed by Comparative Fit Index (CFI), Tucker-Lewis Index (TLI), Root Mean Square Error of Approximation (RMSEA), and Standardized Root Mean Square Residual (SRMR).

To analyze the structural model, we tested our hypotheses in the SEM using maximum-likelihood (ML) estimation. We tested mediating effects of growth language mindset and language learning motivation on relationship between communication anxiety and L2 WTC by computing bias-corrected 95% confidence intervals for 5000 bootstrapped samples using Mplus 8.3. Statistical significance was determined by the non-inclusion of zero in the 95% confidence intervals [45].

## Results

### Descriptive analysis

Using SPSS 26, we conducted an analysis encompassing data screening for missing data and outliers, calculation of descriptive statistics such as means and standard deviations, assessment of normality, and determination of the Pearson correlations between the factors.

To investigate missing data and detect outliers, we conducted a series of data screening procedures. Data sets that contained 80% missing values and values that were considered outliers were eliminated. As a result, 26 cases were found to have problematic values and were subsequently excluded from the original sample of 873 learners, leaving a total of 847 valid cases that were deemed suitable for further analysis.

Among all the variables examined, growth language mindset exhibited the highest mean score ($M = 4.150$, $SD = 1.029$), with language learning motivation ($M = 3.321$, $SD = 1.050$) and L2 WTC ($M = 2.886$, $SD = 1.286$) following closely behind. Notably, communication anxiety displayed the lowest mean score ($M = 2.509$, $SD = 1.123$) (Table 1).

To evaluate the normality of the data, we performed an analysis on the skewness and kurtosis values. According to West et al. [46], a skewness value less than |2| and a kurtosis value less than |7| suggest no significant deviation from normality. It was observed that all the values of the data were within the anticipated range, indicating a normal distribution (Table 1).

**Table 1. Descriptive statistics, normality, and correlations between studied variables.**

|  | M | SD | skewness | kurtosis | CA | GLM | LLM | WTC |
|---|---|---|---|---|---|---|---|---|
| CA | 2.509 | 1.123 | 0.507 | −0.634 | 1 |  |  |  |
| GLM | 4.150 | 1.029 | −1.234 | 0.896 | −0.234** | 1 |  |  |
| LLM | 3.321 | 1.050 | −0.119 | −0.774 | −0.235** | 0.646** | 1 |  |
| WTC | 2.886 | 1.286 | 0.221 | −1.072 | −0.275** | 0.482** | 0.784** | 1 |

Note. M = mean; SD = standard deviation; CA = communication anxiety; GLM = growth language mindset; LLM = language learning motivation; WTC = willingness to communicate

**$p < 0.01$

There are significant correlations between communication anxiety, growth language mindset, language learning motivation, and L2 WTC. To be specific, communication anxiety was negatively correlated with growth language mindset ($r_{(847)} = -0.234$, $p < 0.01$), language learning motivation ($r_{(847)} = -0.235$, $p < 0.01$), and L2 WTC ($r_{(847)} = -0.275$, $p < 0.01$). Growth language mindset was positively correlated with language learning motivation ($r_{(847)} = 0.646$, $p < 0.01$) and L2 WTC ($r_{(847)} = 0.482$, $p < 0.01$). In addition, language learning motivation was positively correlated with L2 WTC ($r_{(847)} = 0.784$, $p < 0.01$). It was determined that there was no issue of multicollinearity as the correlation values were observed to be less than 0.85 [47] (see Table 1). Additionally, this study found no significant association between participants' gender and years of learning English and their L2 WTC. According to Cohen et al. [48], the correlation between two variables can be classified as having a minor, moderate, or main effect size when the values are 0.1, 0.3, and 0.5, respectively. This study found that there were weak correlations between the age ($r_{(847)} = -0.073$, $p < 0.05$) and grade ($r_{(847)} = -0.091$, $p < 0.01$) of participants and their L2 WTC. Therefore, the participants' gender, age, grade, and years of learning English were not included in the major analysis.

## Measurement model assessment

The measurement model is assessed in terms of individual item reliability, internal consistency reliability, convergent validity, and discriminant validity (Table 2).

The individual item reliability was assessed based on the factor loadings of each item on its corresponding construct, where values greater than 0.60 are considered adequate. The results of this study indicate that all items exhibit factor loadings surpassing the 0.60 benchmark, thereby confirming satisfactory reliability of the items within the measurement model, as detailed in Table 2.

To evaluate the internal consistency reliability of the measurement model, we calculated Cronbach's alpha (α) and composite reliability (CR). The commonly accepted criteria for

**Table 2. Results of measurement model assessment.**

| Constructs | CA | GLM | LLM | WTC | Factor loadings | α | CR | AVE |
|---|---|---|---|---|---|---|---|---|
| CA | 0.753 |  |  |  | 0.615–0.811 | 0.920 | 0.921 | 0.567 |
| GLM | −0.234 | 0.899 |  |  | 0.829–0.958 | 0.942 | 0.944 | 0.809 |
| LLM | −0.235 | 0.646 | 0.782 |  | 0.639–0.863 | 0.913 | 0.916 | 0.611 |
| WTC | −0.275 | 0.482 | 0.784 | 0.843 | 0.681–0.924 | 0.921 | 0.924 | 0.710 |

Note. CA = communication anxiety; GLM = growth language mindset; LLM = language learning motivation; WTC = willingness to communicate; α = Cronbach's alpha; CR = composite reliability; AVE = average variance extracted; Diagonal (in bold) represents the square root of the AVE

assessing internal consistency reliability by using Cronbach's alpha are as follows: $\alpha \geq 0.9$ is considered excellent internal consistency reliability, $0.7 \leq \alpha < 0.9$ is good, $0.6 \leq \alpha < 0.7$ is acceptable, $0.5 \leq \alpha < 0.6$ is poor, and $<0.5$ is unacceptable [49]. The Cronbach's alpha for communication anxiety, growth language mindset, language learning motivation and L2 WTC in the study were 0.920, 0.942, 0.913, and 0.921 respectively.

The assessment of CR adhered to the criteria outlined by Hair et al. [50], which stipulate that a score below 0.6 indicates low reliability, while a score between 0.6 and 0.7 is acceptable. A score ranging from 0.7 to 0.9 is considered satisfactory, and a value exceeding 0.95 poses a risk of multicollinearity. Based on these standards, internal consistency reliability of the measurement model is at a satisfactory level (Table 2).

The average variance extracted (AVE) is a reliable assessment for convergent validity, with a minimum threshold of 0.5 [51]. The AVE value of the measurement model surpassed the prescribed threshold, indicating that the measurement model exhibited an acceptable level of convergent validity, as demonstrated in Table 2.

To assess the degree of alignment between empirical data and the proposed model (i.e., model fit), a suite of fit indices was employed, comprising the Comparative Fit Index (CFI), Tucker-Lewis Index (TLI), Standardized Root Mean Square Residual (SRMR), and Root Mean Square Error of Approximation (RMSEA). For CFI and TLI, a value of 0.90 or above is generally considered to indicate a satisfactory fit, while a value of 0.95 or above suggests an excellent fit. For RMSEA, a value of 0.08 or less is typically considered to indicate a satisfactory fit, and a value of 0.06 or less suggests an excellent fit. Similarly, for SRMR, a value of 0.08 or less is generally considered to indicate a satisfactory fit, and a value of 0.06 or less suggests an excellent fit [52]. The findings of this study indicate that the hypothesized model satisfies the criteria for a good fit, with CFI = 0.957, TLI = 0.952, RMSEA = 0.056, and SRMR = 0.051.

## Structural model analysis

Subsequent to the evaluation of the measurement model, we proceeded to investigate the structural model, which sought to establish the relationship between communication anxiety and L2 WTC, with growth language mindset and language learning motivation serving as mediators. Our analysis involved an assessment of the path coefficients in terms of their significance and effect size, as well as an exploration of the direct and indirect effects (Fig 2).

The results indicated that communication anxiety had a significant negative impact on L2 WTC of Chinese middle school students ($\beta = -0.105$, $p < 0.001$), which supports H1. However, H2 was rejected as growth language mindset was found to have a significant negative effect on L2 WTC ($\beta = -0.161$, $p < 0.001$). H3 was confirmed as language learning motivation had a significant positive influence on L2 WTC ($\beta = 1.040$, $p < 0.001$). Additionally, H4 was supported as communication anxiety had a significant negative effect on growth language mindset ($\beta = -0.254$, $p < 0.001$). H5 was also supported as communication anxiety had a significant negative impact on language learning motivation ($\beta = -0.089$, $p < 0.01$) (See Fig 2 and Table 3).

Effect sizes in the structural model were assessed using Cohen et al.'s [48] equation for effect size using beta ($\beta$) values, which were grouped into four categories: 0.0 to 0.1 denoting a weak effect, 0.1 to 0.3 denoting a modest effect, 0.3 to 0.5 denoting a moderate effect, and values exceeding 0.5 denoting a strong effect. According to these criteria, language learning motivation had the most direct positive effect on learner L2 WTC, with a very strong size. Meanwhile, growth language mindset and communication anxiety had modest negative effects on L2 WTC. Furthermore, communication anxiety had a modest negative influence on growth language mindset and a weak negative effect on language learning motivation. In addition,

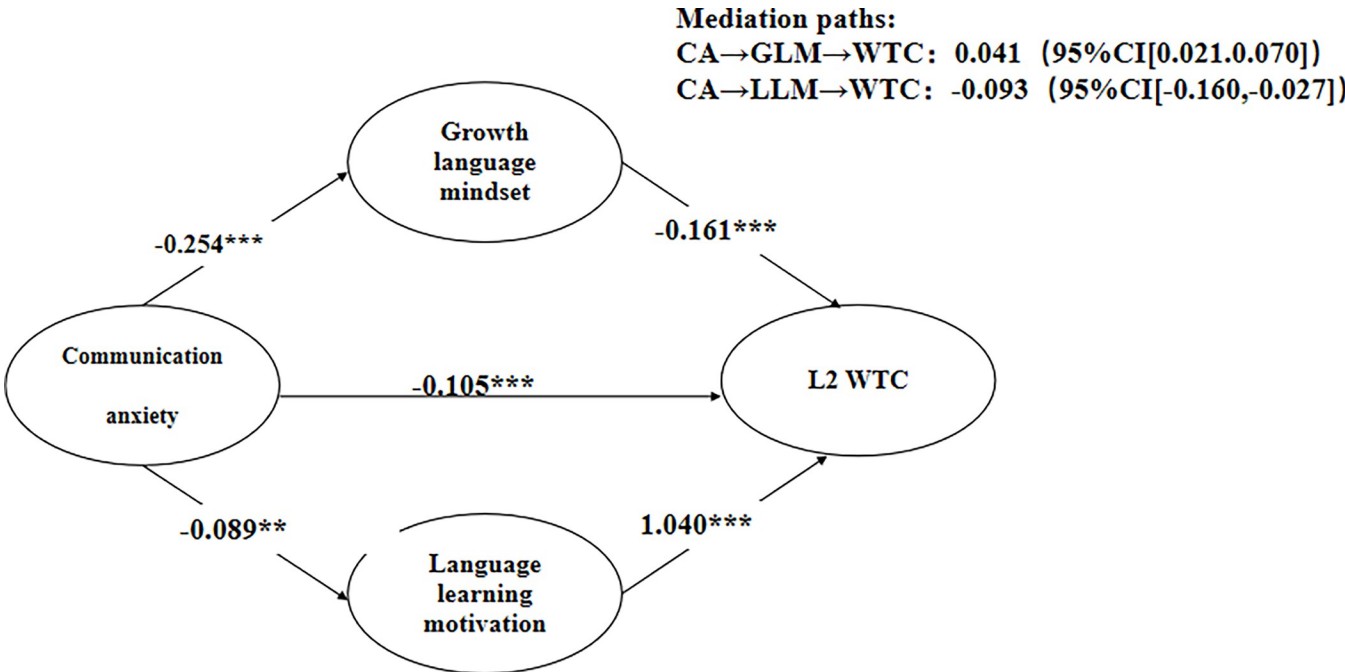

**Fig 2. Structural model of communication anxiety, growth language mindset, language learning motivation, and L2 WTC of Chinese middle school students.** Note: CA = communication anxiety; GLM = growth language mindset; LLM = language learning motivation; WTC = willingness to communicate; N = 847; Survey design-based non-standardized coefficients; Standard errors in parentheses; **p < 0.01, ***p < 0.001.

growth language mindset exerted a strong positive effect on language learning motivation (see Fig 2 and Table 3).

To explore the mediating effects of growth language mindset and language learning motivation on the relationship between communication anxiety and L2 WTC, we performed mediation tests by employing the bootstrapping method, which is a more suitable option than Sobel Test of mediation. The bootstrapping process involved bias-corrected bootstrapping with 5000 iterations and a 95% confidence interval. The mediating effects would be considered significant only if the 95% confidence interval does not contain zero [45].

Table 4 depicts the two significant mediating pathways. Communication anxiety exerts its influence on L2 WTC through two distinct indirect routes, involving either the growth language mindset or language learning motivation.

**Table 3. Direct relationships between communication anxiety, growth language mindset, language learning motivation, and L2 WTC.**

| Hypotheses | Estimate | S.E. | Est./S.E. | Two-Tailed P-Value | Results |
|---|---|---|---|---|---|
| CA→WTC (H1) | −0.105 | 0.029 | −3.681 | 0.001 | Supported |
| GLM→WTC (H2) | −0.161 | 0.039 | −4.165 | 0.001 | Not supported |
| LLM→WTC (H3) | 1.040 | 0.069 | 15.151 | 0.001 | Supported |
| CA→GLM (H4) | −0.254 | 0.044 | −5.757 | 0.001 | Supported |
| CA→LLM (H5) | −0.089 | 0.032 | −2.757 | 0.006 | Supported |

Note. CA = communication anxiety; WTC = willingness to communicate; GLM = growth language mindset; LLM = language learning motivation

Table 4. Mediating effects of growth language mindset and language learning motivation on the relationship between communication anxiety and L2 WTC.

| Path | Point estimate | Product of coefficients | | | Bootstrap 5000 times 95% CI | |
|---|---|---|---|---|---|---|
| | | | | | Bias corrected | |
| | | S.E. | Est./S.E | P | Lower | Upper |
| CA→GLM→WTC (H6) | 0.041 | 0.012 | 3.294 | 0.001 | 0.021 | 0.070 |
| CA→LLM→WTC (H7) | −0.093 | 0.034 | −2.733 | 0.006 | −0.160 | −0.027 |

Note. CA = communication anxiety; GLM = growth language mindset; LLM = language learning motivation; WTC = willingness to communicate

For H6, we proposed that the growth language mindset played a mediating role in the relationship between communication anxiety and L2 WTC. The results of the mediation analysis showed that the indirect negative impact of communication anxiety on L2 WTC, through growth language mindset, is statistically significant albeit weak ($\beta$ = 0.041, 95% CI [0.021, 0.070]), thereby verifying H6.

For H7, we postulated that language learning motivation served as a mediator in the relationship between communication anxiety and L2 WTC. The results indicated that communication anxiety significantly and negatively influenced L2 WTC indirectly through the mediation of language learning motivation. The indirect effects of communication anxiety on L2 WTC through language learning motivation were weak yet statistically significant ($\beta$ = −0.093, 95% CI [−0.160, −0.027]), thus supporting H7.

The findings from the analysis of the structural model (depicted in Fig 2, and presented in Tables 3 and 4) revealed that four out of five hypotheses pertaining to the direct effects were confirmed, and two hypotheses concerning the mediating effects were verified. This implies that 6 out of 7 hypotheses were supported. Moreover, the results of the structural model analysis indicated that the hypothesized model had a noteworthy explanatory power of 64.2% for L2 WTC, indicating a robust explanatory capacity of the proposed model for the outcome variable of L2 WTC.

## Discussion

This study was conducted to examine the interplay between communication anxiety, growth language mindset, language learning motivation, and willingness to communicate in English among middle school students in China. The results of the investigation showed that students' willingness to communicate was impacted directly or indirectly by individual difference variables, including communication anxiety, growth language mindset, and language learning motivation.

### The negative impact of communication anxiety on L2 WTC

The result of this study showed that communication anxiety negatively predicted L2 WTC of Chinese middle school students (Confirming H1), which aligns with a substantial body of research highlighting the detrimental effects of communication anxiety on reduced language use [29–31], indicating that individuals who experienced high levels of communication anxiety were less likely to engage in L2 oral communication. When faced with anxiety, some individuals might resort to avoidance as a coping mechanism, which involves refraining from verbal communication. As anxiety increases, so does the reluctance to communicate, thereby limiting the opportunities for language practice and skill enhancement. The impediment of L2 WTC due to communication anxiety might be attributed to the fear of receiving negative feedback [53], the risk of being ridiculed, and the perceived incompetence [54]. Furthermore, apart

from affecting students' WTC, anxiety has been recognized as a hindrance that disrupts the cognitive ability to perform task-related functions by interfering with the working memory capacity [55].

It is essential that teachers prioritize the implementation of strategies aimed at mitigating communication anxiety. Such strategies include the provision of constructive feedback, the offering of praise, and the facilitation of collaborative learning environments to reduce communication anxiety [56].

## The negative impact of growth language mindset on L2 WTC

The hypothesis that growth language mindset would have a positive impact on L2 WTC in the present study was not supported (refusing H2). The results indicated the opposite: a significant negative influence of growth language mindset on L2 WTC. This finding is inconsistent with Zarrinabadi et al. [15] who discovered a positive association between growth language mindset and L2 WTC. Furthermore, our result is also inconsistent with Wang et al. [20] who found that language mindset did not have a direct effect on L2 WTC in class rather it had an indirect effect on L2 WTC via positive emotions like enjoyment and pride and negative emotion boredom. The negative association between growth language mindset and L2 WTC presents a perplexing issue, which may be attributed to various factors. One possible explanation for this association is the prevalence of exam-driven approaches to English language teaching and learning in China's secondary education [57]. Both instructors and students tend to prioritize receptive skills, such as listening comprehension, reading comprehension over writing and speaking. Consequently, students who possess growth language mindset may dedicate a significant amount of time towards improving their receptive language skills, but may not necessarily focus on improving their speaking abilities. Peng [13] suggest that the Chinese cultural heritage exerts an impact on students WTC. Deeply rooted in Confucianism, the educational philosophy in China emphasizes the cultivation of modesty and humility as core values in the educational process and places a higher value on humility over inherent talent or potential for growth. Ma et al. [27] extended this perspective by noting that students display a significant concern for their social image as perceived by their peers, which can significantly influence their engagement in communicative activities. Therefore, as Lou & Li [58] have posited that growth mindset may not yield the desired outcomes or may even have negative consequences in societies where fixed-mindset norms are prevalent. Therefore, under the above-mentioned circumstances, a growth mindset might not translate into increased WTC.

## The positive impact of language learning motivation on L2 WTC

The present study revealed that language learning motivation had a positive impact on L2 WTC among middle school students in China (verifying H3). This result is consistent with prior research [12, 35, 36]. Notably, the study identified language learning motivation as the most significant factor that directly influenced L2 WTC among Chinese middle school students. This underscores the pivotal role of motivation in shaping learners' L2 WTC. Yashima [12] found a direct positive link between motivation and L2 WTC, which implies Learners with high level of motivation in learning a language are more likely to align their goals with the attributes associated with using L2 fluently [13], which enhances their WTC in that language. The meta-analysis by Shirvan et al. [35] further substantiated this link, with a substantial correlation between language learning motivation and L2 WTC ($r = 0.370$ and CI = [0.32 to 0.42]). Additionally, our study is consistent with Alrabai's [36] findings, confirming that language learning motivation significantly and directly influences L2 WTC among EFL learners.

This alignment with previous research underscores the enduring and foundational role of motivation in the communicative language learning process.

## The negative impact of communication anxiety on growth language mindset

The results of the present study confirmed that communication anxiety had a negative effect on growth language mindset (supporting H4), which is consistent with previous research [9]. Our study has demonstrated that anxiety and growth mindset are inversely related in educational settings. The negative relationship between these two variables can be attributed to the fact that individuals who experience communication anxiety may be more sensitive to social rejection [9], lack confidence in their L2 abilities, encounter negative social interactions, and perform poorly in language tasks. As a result, these unfavorable experiences may further reinforce the students' negative and fixed perception of their L2 proficiency, leading them to adopt fixed mindset instead of growth language mindset. This negative emotion not only negatively affects growth language mindset, but also has an adverse effect on WTC. Therefore, language instructors should quip students with anxiety-reducing techniques and encourage students to recognize their progress and praise progress made to boost learners' confidence. By implementing these strategies, instructors may help learners minimize the negative impact of communication anxiety and cultivate a growth language mindset.

## The negative impact of communication anxiety on language learning motivation

The results of this study indicated that communication anxiety had a negative impact on language learning motivation (confirming H5), which is consistent with previous research [38–40]. According to Gardner's socio-educational model of second language acquisition, Bernaus et al. [59] argued that language learning anxiety was negatively associated with language learning motivation. Anxiety had negative effects, hindering the L2 learning process by undermining the positive effects of learners' motivation. Conversely, lower levels of anxiety may lead to higher achievement in the L2, as learners with lower anxiety levels tend to possess greater self-confidence and higher levels of motivation in their language learning. Therefore, instructors can facilitate higher levels of language learning motivation by creating positive and enjoyable learning experiences and supportive classroom environment that are free from anxiety, which in turn can lead to a greater L2 WTC [60]. Thankfully, the current study demonstrated that the influence of communication anxiety on language learning motivation is minimal, indicating a relatively favorable result.

## Growth language mindset as a mediator between communication anxiety and L2 WTC

The present study uncovered that growth language mindset acted as a mediator between communication anxiety and L2 WTC among Chinese middle school students (confirming H6). Our finding partly echoes with Zarrinabadi et al. [15], which indicated that growth language mindset can either separately or jointly with communicative competence mediates the relationship between autonomy support and WTC.

Individuals who suffer from communication anxiety may be more sensitive to social rejection, exhibiting reduced confidence in their L2 competencies, encountering negative social interactions, and performing poorly in linguistic tasks. As a result, such adverse experiences may further reinforce the students' negative and fixed perception of their L2 proficiency,

resulting in the adoption of fixed language mindset, while learners who endorse growth language mindset are inclined towards improvement and advancement.

However, within the context of an exam-driven Asian foreign language learning environment, the predominant focus is on achieving high grades by excelling in vocabulary, reading comprehension, and writing [24]. Consequently, learners tend to allocate minimal effort to enhancing their oral communication skills, which accounts for the observed deficiency in their L2 WTC.

## Language learning motivation as a mediator between communication anxiety and L2 WTC

The present study demonstrated that language learning motivation functioned as a mediator in the association between communication anxiety and L2 WTC among Chinese middle school students (confirming H7). This outcome partly aligns with previous research findings [36], which has found motivation acting as a mediator between grit and L2 WTC. However, our finding of communication anxiety influencing L2 WTC indirectly through learner motivation different from the pathways identified by Yu [61], where L2 motivation influenced L2 WTC via L2 anxiety. These findings do not conflict but rather reinforce the bidirectional nature of the relationship between negative emotions and motivation. Moreover, learner affect may serve as a predictor and/or mediator in various communication models.

Higher levels of communication anxiety can contribute to lower language learning motivation and subsequently result in lower L2 WTC. Conversely, lower levels of communication anxiety are associated with higher language learning motivation, which eventually leads to higher WTC. These findings have demonstrated that a heightened level of motivation to acquire a language can mitigate the adverse effects of communication anxiety, thereby promoting an enhanced inclination to communicate in L2.

## Theoretical implications and pedagogical implications

In this study, we integrated multiple predictors of L2 WTC: communication anxiety (an affective variable), language learning motivation (a cognitive driver), and the growth language mindset (a belief variable). This comprehensive model extends prior research by encompassing a triadic interplay of these factors, enhancing the predictive power.

Additionally, the study's findings highlight the mediating roles of growth language mindset and language learning motivation in the relationship between communication anxiety and L2 WTC. The result demonstrated that communication anxiety had a negative impact on L2 WTC of Chinese middle school students. To reduce anxiety, it is recommended that strategies like providing positive feedback, offering praise and incorporating collaborative work or task-based learning by fostering a supportive classroom atmosphere [56].

One noteworthy finding is the negative effect of growth language mindset on L2 WTC. This finding challenges the idea that growth mindset has ubiquitous positive impacts in most learning contexts. It aligns with Jiang et al. [62] observation of a weak predictive effect of language mindsets on learning outcomes among Chinese young learners. Educators are advised to foster autonomy in students, encouraging them to view communication challenges as opportunities for growth rather than as threats to self-esteem. In east-Asian Confucian cultures, teachers should nurture students' risk-taking abilities and recognize the impact of a growth language mindset may vary across different learning stages, necessitating tailored teaching strategies according to the learners' characteristics.

Another noteworthy finding is the substantial positive influence of language learning motivation on L2 WTC, as well as its mediating role between communication anxiety and L2

WTC. This highlights the crucial role of motivation in promoting learners' L2 WTC. In the digital era, educators are recommended to adopt various measures to boost learners' motivation, such as digital games [63], digital storytelling [64], and online cooperative learning [65]. Additionally, innovative educational technologies, including artificial intelligence (AI), augmented reality (AR), and virtual reality (VR) can also be to leveraged to enhance students' engagement, and strengthen their language learning motivation [66, 67], thereby advancing their proficiency in L2 and enabling learners to attain their language learning objectives.

### Limitations and future directions

While the present research offers significant contributions, it is not entirely exempt from limitations. To begin with, it would be advantageous to duplicate the study in other environments, including high schools, universities, and international contexts, to evaluate the consistency of the findings. In addition, the cross-sectional data serves as an initial foundation for investigating the complex interrelationships among communication anxiety, growth language mindset, language learning motivation, and L2 WTC. To ascertain the exact causal direction of these effects, it would be beneficial to gather longitudinal data. Lastly, the current study was carried out utilizing a quantitative methodology and solely relied on self-reported surveys for data collection. As a result, it is recommended that forthcoming research incorporate a triangulation of diverse data collection methods, comprising of interviews and observations.

### Conclusion

In conclusion, this research provides a refined understanding of the dynamics between communication anxiety and L2 WTC, with growth language mindset and language learning motivation acting as mediators. Communication anxiety has been found to exert adverse effects on L2 WTC, growth language mindset, and language learning motivation respectively. Notably, apart from being a negative predictor of L2 WTC, growth language mindset also mediates the impact of communication anxiety on L2 WTC. Additionally, a positive correlation was established between language learning motivation and L2 WTC. Language learning motivation was also identified as a mediator between communication anxiety and L2 WTC, suggesting that fostering language learning motivation could be a strategic approach to mitigate the negative effects of anxiety onL2 WTC. These findings underscore the complex interplay among these factors and highlight potential areas for intervention to boost L2 WTC and optimize second language learning outcomes.

### Supporting information

**S1 File. Survey items.**
(DOCX)

**S1 Dataset. Dataset collected from the survey.**
(XLS)

### Author Contributions

**Conceptualization:** Juan Wang.

**Data curation:** Juan Wang, Cunying Fan.

**Methodology:** Juan Wang.

**Resources:** Tongquan Zhou.

**Software:** Cunying Fan.

**Supervision:** Tongquan Zhou.

**Validation:** Cunying Fan.

**Visualization:** Cunying Fan.

**Writing – original draft:** Juan Wang.

**Writing – review & editing:** Tongquan Zhou, Cunying Fan.

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
