## [Decision Letter · Decision Letter 0]

30 Aug 2024

PONE-D-24-19380Impact of communication anxiety on L2 WTC of middle school students: Mediating effects of growth language mindset and language learning motivationPLOS ONE

Dear Dr. Wang,

Thank you for submitting your manuscript to PLOS ONE. After careful consideration, we feel that it has merit but does not fully meet yet PLOS ONE’s publication criteria as it currently stands. Therefore, we invite you to submit a revised version of the manuscript that addresses the points raised during the review process. As you will see, both reviewers kindly provided a very accurate and detailed list of points that should be carefully addressed. Most of those are minor edit styles but the question about Ethics approval raised by one of the reviewers and the need for framing the current results of this manuscript against past empirical findings should require more attention, hence the major revision. Nonetheless, I agree with the reviewers in that the manuscript is currently interesting and it can be a valuable contribution for the readership of this journal venue.

We look forward to receiving your revised manuscript.

Kind regards,

Massimo Stella, PhD

Academic Editor

PLOS ONE

Journal requirements: 1. When submitting your revision, we need you to address these additional requirements. Please ensure that your manuscript meets PLOS ONE's style requirements, including those for file naming. The PLOS ONE style templates can be found at https://journals.plos.org/plosone/s/file?id=wjVg/PLOSOne_formatting_sample_main_body.pdf and https://journals.plos.org/plosone/s/file?id=ba62/PLOSOne_formatting_sample_title_authors_affiliations.pdf. 2. We note that the grant information you provided in the ‘Funding Information’ and ‘Financial Disclosure’ sections do not match.  When you resubmit, please ensure that you provide the correct grant numbers for the awards you received for your study in the ‘Funding Information’ section. 3. [The authors would like to thank all students for their time and willingness to participate in the survey. We also would like to thank the Teaching Reform Program (Number:22jg41) of XX University.]We note that you have provided funding information that is not currently declared in your Funding Statement. However, funding information should not appear in the Acknowledgments section or other areas of your manuscript. We will only publish funding information present in the Funding Statement section of the online submission form. Please remove any funding-related text from the manuscript and let us know how you would like to update your Funding Statement. Currently, your Funding Statement reads as follows:  [The author(s) received no specific funding for this work.] Please include your amended statements within your cover letter; we will change the online submission form on your behalf. 4. We note that your Data Availability Statement is currently as follows: [All relevant data are within the manuscript and its supporting information files.] Please confirm at this time whether or not your submission contains all raw data required to replicate the results of your study. Authors must share the “minimal data set” for their submission. PLOS defines the minimal data set to consist of the data required to replicate all study findings reported in the article, as well as related metadata and methods (https://journals.plos.org/plosone/s/data-availability#loc-minimal-data-set-definition). For example, authors should submit the following data: - The values behind the means, standard deviations and other measures reported;- The values used to build graphs;- The points extracted from images for analysis. Authors do not need to submit their entire data set if only a portion of the data was used in the reported study. If your submission does not contain these data, please either upload them as Supporting Information files or deposit them to a stable, public repository and provide us with the relevant URLs, DOIs, or accession numbers. For a list of recommended repositories, please see https://journals.plos.org/plosone/s/recommended-repositories. If there are ethical or legal restrictions on sharing a de-identified data set, please explain them in detail (e.g., data contain potentially sensitive information, data are owned by a third-party organization, etc.) and who has imposed them (e.g., an ethics committee). Please also provide contact information for a data access committee, ethics committee, or other institutional body to which data requests may be sent. If data are owned by a third party, please indicate how others may request data access. 5. Please include your full ethics statement in the ‘Methods’ section of your manuscript file. In your statement, please include the full name of the IRB or ethics committee who approved or waived your study, as well as whether or not you obtained informed written or verbal consent. If consent was waived for your study, please include this information in your statement as well.  

Reviewers' comments:

Reviewer's Responses to Questions

**Comments to the Author**

1. Is the manuscript technically sound, and do the data support the conclusions?

Reviewer #1: Yes

Reviewer #2: Yes

2. Has the statistical analysis been performed appropriately and rigorously? 

Reviewer #1: Yes

Reviewer #2: Yes

3. Have the authors made all data underlying the findings in their manuscript fully available?

Reviewer #1: Yes

Reviewer #2: No

4. Is the manuscript presented in an intelligible fashion and written in standard English?

Reviewer #1: No

Reviewer #2: No

5. Review Comments to the Author

Reviewer #1: This article investigates the relationship between communication anxiety (CA) and willingness to communicate (WTC) as mediated by the role of language mindset (GLM) and language learning motivation (LLM). Self-report measures were administered to Chinese students learning English as a second language. Structural equation modeling (SEM) was employed to test the parallel mediation hypothesis. All direct and indirect paths were examined. All hypotheses but one were supported (i.e., the relationship between GLM and WTC).

This article is very interesting and provides insights into the mechanisms through which anxiety, motivation, and mindset can influence willingness to communicate. The introductory section is sufficiently detailed, clearly explaining the constructs involved in the analyses and the hypothesized relationship. The Discussion and Conclusion sections reflect the results found in the analyses, and the limitations are complete and address the challenges that can be encountered in conducting this type of study, such as cross-sectional and self-report data. In general, this study may add significant value to the literature on second language acquisition, by unravelling the complex relationship between CA, WTC, GLM, and LLM. However, I must highlight a series of points that concern, on one hand, a need for greater clarity in some crucial areas and, on the other hand, a request for greater attention to writing style. Hereafter, I will present my observations.

Major points:

• Page 31, lines 627-633: In these lines, it is mentioned that the current study found that anxiety indirectly affects WTC in L2 through student motivation, which contrasts with the paths identified by Khajavy et al. (2016), who hypothesized a bidirectional role of anxiety with motivation. Perhaps adding a few lines explaining more clearly why this bidirectionality was hypothesized could further strengthen your results.

• Page 23-24, lines 467, 485-488: Nine hypotheses and three indirect paths are mentioned. However, according to the introduction, the number of hypotheses should be 7 and the number of hypothesized indirect paths should be 2 (also looking at Table 3). Hence, there is an inconsistency that deserves to be corrected or explained.

• The manuscript would benefit from an English language revision (see below).

Minor points:

• The acronym “SLA” is defined on page 6, line 128. However, it is used earlier on page 3, line 59. Defining it earlier in the paper would aid comprehension.

• For p-values: replace `.000` with `< .001`.

There are several typos in the manuscript concerning punctuation, spacing, and verb tenses. Below are some examples; there are many more throughout the manuscript. Therefore, I suggest a complete grammatical and linguistic revision of the document.

• Page 14, line 306: “the” is repeated twice.

Punctuation.

• Page 27, line 555: mindset.

• Page 28, line 572: motivation.

Spacing between words. Generally, there are many omissions of spaces throughout the manuscript, especially in citations. Below are just a few examples.

• Page 3, line 51: “performance[4]” should be “performance [4]”

• Page 3, line 65: “emotions,mindset and WTC[18]; the” should be “emotions, mindset and WTC [18]; the”

Grammatical errors:

• Page 4, line 72: “represent” should be “represents”

• Page 10, line 201: “arise” should be “arises”

• Page 20, lines 415-416: “A excellent” should be “An excellent”

Reviewer #2: This manuscript reports an interesting and well conceived study investigating the relationship between communication anxiety and willingness to communicate in a foreign language and the potential mediating roles of learner motivation and growth language mindset. Overall, this appears to be a rigorous study and the conclusions drawn from the data are warranted. However, there are some areas to be clarified and improved, including the writing. Please find my detailed comments below.

Research ethics – the ethics statement suggests the manuscript rather than the study was reviewed by the relevant ethics committee of the authors’ institution. Moreover, the ethics statement does not report the outcome of the review (e.g., approval). Can the authors clarify whether they sought ethical approval for the study prior to data collection or if they sought ethical approval only on the manuscript after the study was conducted? If ethical approval was sought for the study, then the ethics statement should be updated. If ethical approval was not sought for the study, then the authors must provide a detailed statement explaining why it was not needed. Information about obtaining informed consent is missing from the ethics statement. Information about the ethics review is missing from the manuscript. See https://journals.plos.org/plosone/s/submission-guidelines#loc-guidelines-for-specific-study-types for further information.

Data availability – the authors have stated that the data is available in the manuscript and supporting documents, but a link to where the data set is stored has not been provided. It may be that the authors have not understood the requirement to provide a minimal data set behind the results reported in the manuscript. The authors should refer to the following guidelines https://journals.plos.org/plosone/s/data-availability

Writing – the manuscript would benefit from copyediting as there were grammatical errors throughout the manuscript and sentences / sections that could benefit from better organisation. Moreover, the typesetting needs further work e.g., spacing between the words, references and punctuation needs reviewing. I have provided some examples in my comments below.

Introduction

Note that numbers refer to manuscript line numbers.

40 – 43 tense of the sentence is not quite correct. Suggest: ‘They proposed that L2 WTC involved learners’ readiness to engage in discourse at a specific time with certain individuals using L2, representing the final psychological stage before actual communication behaviour’. It is also not exactly clear how “and was a crucial objective of L2 education” fits with the rest of the sentence – I would suggest removing it.

50 – 51 – Does communication anxiety always lead to a decline in performance? If not, then I suggest rewording to ‘and may lead to a decline in language performance’.

60 – Note that the noun ‘affect’ cannot be referred to in plural as ‘affects’ (affects with an ‘s’ is the verb form, with a different meaning). The sentence should be edited to ‘…especially positive emotions and affect play on L2 WTC…’.

67 – I believe the authors mean ‘university students’ rather than ‘universities students’

66 – ‘school’ missing i.e., ‘middle school students’

74 – 77 – The use of the term ‘examinations’ here is not clear, as many language examinations include the assessment of spoken communication skills. Perhaps the authors mean examinations focused on receptive skills (e.g., reading and listening) or skills other than spoken communication?

83 – 92, I would suggest condensing the potential implications of the study in the introduction as the introduction section is quite long given the literature review that follows.

95 – 99 – citations are needed for the statements opening this paragraph.

109 – a definition / further explanation of ‘ideal L2 selves’ is needed here.

124 – it is not clear what is meant by the term ‘research subjects’ here as it could refer to research participants or research topics. Suggest revising the sentence from lines 123- 125 for clarity.

133- 136 – This sentence does not currently make grammatical sense. I believe the word ‘under’ has been included by mistake, and ‘and’ is missing before ‘whether’ (which is currently misspelled). Please review this sentence and revise.

148 (and lines 180, 197, 219, 237) the word ‘predicts’ with an ‘s’ should be used in all hypotheses in which it appears.

164 – ‘responsibility’ not ‘responsibilities’

165 – this sentence is worded in a way that suggests a causal relationship between L2 growth language mindset and WTC, however it’s not clear whether the methodology of the research described thereafter allows for causal relationships to be determined. Suggest rewording to ‘Recent research suggests that growth language mindset has a positive impact on…’

200 – 204 – Communication anxiety has already been defined/ described earlier in this paper and these sentences are repetitive. Suggest starting this section from ‘Individuals who suffer…’.

208 - Similar to previous comment, fixed mindset has been described earlier in the introduction and is not needed here.

211- Not clear how the Noels [9] study relates to GLM. The authors should make this clear in their description of the study.

Methods

267 – ‘22.4% had been learning English…’ not ‘has been learning’.

Line 271 – 301. The authors should provide further information about how the validated scales were modified for the current study.

Line 287 – it would be helpful for the reader if ‘Lou et al.’ is also included next to citation [43] (as the authors did for the scale of Horwitz et al. [42] on line 280).

Lines 285, 289, 293, 298 – should say ‘Cronbach’s alpha’ (singular) not ‘alphas’.

291 – place Liu before the numbered citation – ‘Liu [44]’

296 – similar to comment for line 287, it would be helpful for the reader if ‘Baghaei et al.’ is also included next to citation [45].

Lines 281 – 283 and lines 299 – 301 repeats some of the same information. As lines 299 – 301 are a summary of the Likert scales for all the measures, remove the information about the Likert scale from lines 281 – 283.

Results

Line 313 shows an example of no space between a full stop and the start of the next sentence. Line 320 shows another example of this. Please edit the entire manuscript for these typesetting errors.

366 – ‘…motivation was positively correlated…’ not ‘were positively correlated’.

381 – suggest using the word ‘main instead of ‘major’.

414- 419 – the authors describe the fit indices but do not describe whether values should be above or below these - please specify this for full information.

415 – ‘An excellent fit...’ instead of ‘A excellent fit…’

470 – please ensure that tables do not run over two pages

485 – 488 – The number of hypotheses detailed here do not correspond to the number of hypotheses described in the tables or in the introduction. Please check this and revise accordingly.

Discussion

General comments – the authors provide some areas of focus for enhancing positive learning mindsets and emotions in the classroom but often do not go as far as providing suggestions for how teachers can do this. The pedagogical suggestions could be strengthened. There are a few instances noted below.

Authors should also aim to minimise repetition across this section.

499 – I believe this should be a subheading – please format accordingly. The same comment applies for the other subheadings in this section (lines 518, 542, 554, 571, 587, 607).

502 – ‘Our study suggests…’ rather than ‘Our studies suggest’ as the authors have conducted one study only.

506 – 510 – citation(s) should be included to support these statements.

519 – Suggest rewording to ‘The hypothesis that growth language mindset would have a positive impact on L2 WTC in the present study was not supported (refuting H2).’

520 – suggest writing the opposite finding out in full.

523 – 525 – suggest removing the finding about within classroom effects from this sentence to improve the flow of the sentence i.e., ‘…Wang et al [18] who found that growth language mindset had a direct impact on L2 WTC outside the classroom’.

530 – similar to a previous comment (lines 74 -77), clarify if productive skills examinations are meant here as many language examinations also include a spoken communication component. Similarly in relation to line 532 – speaking abilities are ‘language skills’ too, so the authors should clarify if they mean ‘receptive’ language skills here or some other meaning.

533 – 541 – The alternative explanation offered is an interesting one. I feel this section would benefit from an additional sentence after the description of Ma et al (line 539) to explicate that the cultural differences described may therefore impact willingness to communicate as this link is implied but not spelled out. The final sentence of this paragraph is fine as it is.

547 – an example of additional spaces between punctuation. Please check the manuscript thoroughly for such typesetting errors.

547 – Typo, suggest ‘which enhances their willingness to communication in that language’.

568 – The authors should include some suggestions for how teachers can minimise the impact of communication anxiety when teaching English.

581 – Similar to previous comment, the authors should include some suggestions for facilitating language learner motivation/ positive and enjoyable learning experiences.

604 – ‘mediate’ not ‘mediates’.

605 – Typo - two instances of ‘and’ are included here.

650 – rather than the ‘malleability of intelligence’ the sentence should state ‘the malleability of language ability’ as the study specifically focuses on growth language mindset rather than general intelligence.

666 - Authors should include suggestions for strategies to enhance the learner variables discussed.

678 – 683 – citations should be provided to support the claims of technologies enhancing the learner variables discussed.

6. PLOS authors have the option to publish the peer review history of their article (what does this mean?). If published, this will include your full peer review and any attached files.

Reviewer #1: No

Reviewer #2: No

---

## [Author Response · Author response to Decision Letter 0]

10 Oct 2024

Response to the editor

Dear editor,

Thank you for your help in the process of submitting and revising. As requested, a rebuttal letter, the mark-up copy of my manuscript,and an unmarked version of my revised paper are uploaded.Also the ethics statement is included in the 'Methods' section of my manuscript. I also added the funding information in my Cover Letter.

Response to the reviewers

Response to Comments of Reviewer 1 

Major points

1. Page 31, lines 627-633: In these lines, it is mentioned that the current study found that anxiety indirectly affects WTC in L2 through student motivation, which contrasts with the paths identified by Khajavy et al. (2016), who hypothesized a bidirectional role of anxiety with motivation. Perhaps adding a few lines explaining more clearly why this bidirectionality was hypothesized could further strengthen your results.

Response：

We extend our sincere gratitude for the reviewer’s valuable insights, which are greatly acknowledged. Below please find the detailed elaboration from line 759 to line 765 in the Mark-up Version: “However, our finding of communication anxiety influencing L2 WTC indirectly through learner motivation different from the pathways identified by Khajavy et al. [62] and Yu [63], where L2 motivation influenced L2 WTC via L2 anxiety. These findings do not conflict but rather reinforce the bidirectional nature of the relationship between negative emotions and motivation. Moreover, learner affect may serve as a predictor and/or mediator in various communication models”.

2. Page 23-24, lines 467, 485-488: Nine hypotheses and three indirect paths are mentioned. However, according to the introduction, the number of hypotheses should be 7 and the number of hypothesized indirect paths should be 2 (also looking at Table 3). Hence, there is an inconsistency that deserves to be corrected or explained.

Response: 

We concur with the accuracy of your suggestions, and have duly incorporated them. The manuscript has been revised in line with your guidance, with changes clearly annotated. We wish to reiterate our appreciation for your contributions. For details, please see line 579 to 581 in the Mark-up Version.

Minor points

Response: 

Upon careful consideration of the thorough and insightful comments offered by the reviewer, we have implemented the subsequent revisions to address the noted issues within our manuscript. Specifically, we have rectified typographical errors pertaining to punctuation, and spacing, in addition to addressing grammatical inaccuracies. We express our sincere gratitude for the meticulous guidance provided.

Response to Comments of Reviewer 2

Major points

1. Research ethics statement

Response:

We extend our apologies for not including the ethical review report for the project in our previous submission. Due to the necessity for our institutional ethics review board to retain the original copy of the ethical review report for archival purposes, we had not previously made a duplicate of the original ethical review approval form. In compliance with your request, I have now added the copy. It is now furnished as supporting material to the submitted material. We appreciate your valuable suggestions. 

2.Data availability——the authors have stated that the data is available in the manuscript and supporting documents, but a link to where the data set is stored has not been provided. It may be that the authors have not understood the requirement to provide a minimal data set behind the results reported in the manuscript.

Response: 

Thank you so much for your advice. During the online submission process, this component was inadvertently omitted from the initial submission. We deeply regret any inconvenience this may have caused and appreciate your understanding. Now the dataset in question has been uploaded as supporting material.

2.Writing – the manuscript would benefit from copyediting as there were grammatical errors throughout the manuscript and sentences / sections that could benefit from better organisation.

Response: 

We extend our sincere gratitude for the reviewer's diligent and thoughtful guidance pertaining to linguistic and mechanical aspects of our manuscript. Following the meticulous examples offered by the reviewer, we have undertaken a thorough revision of each item, ensuring a comprehensive enhancement of the text. Introduction

Grammatical errors

40-43;60;66;67;124;133-136;148;164;200-204;208

Response: 

The above-mentioned grammatical errors have been corrected.

Word choice

50-51

Response: 

According to the reviewer’s advice, we use the down toner “may” to soft the tone. Please see line 43 in the Mark-up version.

74-77 

Response:

We concur with the reviewer's perspective. We intend to emphasize the examinations that focus on receptive skills, encompassing a broad range of passages designed to assess learners' abilities in listening and reading comprehension. “Both instructors and students tend to prioritize receptive skills, such as listening comprehension, reading comprehension, over writing and speaking.” For details, please see line 72 to 73 in the Mark-up version.

Content

Line 83-92;109;165;211

Response: 

We express our gratitude to the reviewer for the thorough and insightful feedback. We have made the necessary subsequent amendments in accordance with the reviewer's recommendations. For details, please kindly turn to the Mark-up version 

Line 83-92 shortened version

Response: 

We shortened this part into the following sentences (For details, please see Mark-up version line 81 to line 92). “It is expected to provide substantial theoretical and practical implications for augmenting L2 WTC of middle school students. Additionally, it is also expected to highlight the role of learner-related factors in cultivating L2 WTC, offering practical insights for the development of educational strategies aimed at enhancing L2 WTC

Line 109

Response: 

Thank you for the constructive advice. We added further elaboration on ideal L2 Selves in the Mark-up version. Please see line 109 to line 111. “The research demonstrated that ideal L2 selves, which encompassing learners' aspirations and their envisioned future linguistic identities significantly correlates with WTC in English among Japanese high school students, both in and out of the classroom.”

Methods

Line 285;267;287;289;293;298291;296;281-283;299-301

Response: 

Grammatical errors as well as technical problems from the above-mentioned lines have been corrected and solved according to the reviewer’s advice.

Line 271-301

Response: 

Detailed modification to the validated scales were added from line 312 to line 319 in the mark-up version: “(1) Items were rephrased to reflect cultural norms and experiences specific to middle school students;(2) Language were simplified to ensure it is accessible to youngsters; (3) The number of items were reduced as well as structure simplified to reduce the cognitive load. The English questionnaire was translated into Chinese by the first writer and the translated questionnaire items were subsequently submitted to three linguistic experts for validation to ensure their accuracy. Based on the feedback provided by the experts, the language of the translated items underwent further refinement.”

Results

Line 313;366;381;415;

Response: 

Thank you so much for pointing out. The issues pertaining to spacing and language have been meticulously revised based on the reviewer's constructive feedback.

Line 414-419

Response:

Detailed fit indices were added in the Mark-up version from line 499 to line 505. The following are the added information:

“For CFI and TLI, a value of 0.90 or above is generally considered to indicate a satisfactory fit, while a value of 0.95 or above suggests an excellent fit. For RMSEA, a value of 0.08 or less is typically considered to indicate a satisfactory fit, and a value of 0.06 or less suggests an excellent fit. Similarly, for SRMR, a value of 0.08 or less is generally considered to indicate a satisfactory fit, and a value of 0.06 or less suggests an excellent fit [52].”

Line 485-488

Response: 

We have meticulously reviewed the hypotheses and implemented the necessary adjustments as per the reviewer's recommendations. For details, please see the Mark-up version from line 579 to line 581.

Discussion

Line 499;518;542;554;571;587;607

Response

Subheadings were added respectively. Please kindly turn to line 593 to line 594;line 627 to line 629;line 664 to line 666; line 688 to line 690;line 707 to line 709;line 726 to line 727, and line 751 to line 754.

Line 502;519;520;523-525;547;604;605;650

Response:

First and foremost, we express our sincere gratitude for the patient and meticulous suggestions provided by the reviewer. We found the feedback to be highly constructive and have accordingly made the necessary revisions.

Line 533-541

Response:

We thank the reviewer for the advice. We added the following line in the Mark-up version (Line 654 to line 662). “which can significantly influence their engagement in communicative activities. Therefore, as Lou & Li [58] have posited that growth mindset may not yield the desired outcomes or may even have negative consequences in societies where fixed-mindset norms are prevalent. Some cultures may place a higher value on humility and effort rather than on inherent talent or potential for growth. Additionally, a growth mindset may not necessarily enhance WTC in learning contexts that are highly competitive or discourage risk-taking. Thus, under the above-mentioned circumstances, a growth mindset might not translate into increased WTC.”

Line 568;581,666

Response:

We appreciate your valuable feedback. Upon careful consideration, we think that it might be more beneficial to include suggestions for teachers and educators within the section of “Theoretical and pedagogical implications”. Therefore, we have decided to incorporate the “Advice for teachers and educators” within Section 5.8. Please kindly turn to the Mark-up version from line 782 to line 812. 

we included actionable strategies that educators can employ to mitigate communication anxiety and foster greater language learning motivation among their students.

Line 678-683

Response:

Thank you for the advice. The role of technology in enhancing student engagement and boosting language learning motivation is supported by specific literature. Please kindly turn to line 807 to line 812 in the Mark-up version. “Educators are recommended to adopt various measures to boost learners’ motivation, such as digital storytelling [64]; digital games [65]; online cooperative learning [66]. In the digital era, innovative educational technologies, including artificial intelligence (AI), augmented reality (AR), and virtual reality (VR) can be to leveraged to enhance students’ engagement, and strengthen their external and internal motivation [67,68], thereby advancing their proficiency in L2 and enabling learners to attain their language learning objectives.”

---

## [Decision Letter · Decision Letter 1]

5 Dec 2024

Impact of communication anxiety on L2 WTC of middle school students: Mediating effects of growth language mindset and language learning motivation

PONE-D-24-19380R1

Dear Dr. Wang,

We’re pleased to inform you that your manuscript has been judged scientifically suitable for publication and will be formally accepted for publication once it meets all outstanding technical requirements.

Kind regards,

Massimo Stella, PhD

Academic Editor

PLOS ONE

Additional Editor Comments (optional):

Reviewers' comments:

Reviewer's Responses to Questions

**Comments to the Author**

1. If the authors have adequately addressed your comments raised in a previous round of review and you feel that this manuscript is now acceptable for publication, you may indicate that here to bypass the “Comments to the Author” section, enter your conflict of interest statement in the “Confidential to Editor” section, and submit your "Accept" recommendation.

Reviewer #1: All comments have been addressed

Reviewer #2: All comments have been addressed

2. Is the manuscript technically sound, and do the data support the conclusions?

Reviewer #1: Yes

Reviewer #2: Yes

3. Has the statistical analysis been performed appropriately and rigorously? 

Reviewer #1: Yes

Reviewer #2: Yes

4. Have the authors made all data underlying the findings in their manuscript fully available?

Reviewer #1: Yes

Reviewer #2: Yes

5. Is the manuscript presented in an intelligible fashion and written in standard English?

Reviewer #1: Yes

Reviewer #2: Yes

6. Review Comments to the Author

Reviewer #1: I read the new version of the manuscript after this round of revision. I also read responses to my and the other reviewer's requests. The authors properly addressed all the points, hence I have no further observations to do. I congratulate with with the authors for this interesting paper.

Reviewer #2: In addition to the specific comments I had on the manuscript being addressed, the previous question I had raised about research ethics approval has thankfully been answered satisfactorily.

Please do proofread the manuscript once again as there are some small issues in the new text that has been added. However, I feel that the clarity and organisation of the manuscript is greatly improved since the first submission.

7. PLOS authors have the option to publish the peer review history of their article (what does this mean?). If published, this will include your full peer review and any attached files.

Reviewer #1: No

Reviewer #2: No

---

## [Editor Report · Acceptance letter]

3 Jan 2025

PONE-D-24-19380R1 

PLOS ONE

Dear Dr. Wang, 

I'm pleased to inform you that your manuscript has been deemed suitable for publication in PLOS ONE. Congratulations! Your manuscript is now being handed over to our production team.

Kind regards, 

on behalf of

Professor Massimo Stella 

Academic Editor

PLOS ONE